# COVID-19 Pandemic Increases the Impact of Low Back Pain: A Systematic Review and Metanalysis

**DOI:** 10.3390/ijerph19084599

**Published:** 2022-04-11

**Authors:** Giuseppe Francesco Papalia, Giorgia Petrucci, Fabrizio Russo, Luca Ambrosio, Gianluca Vadalà, Sergio Iavicoli, Rocco Papalia, Vincenzo Denaro

**Affiliations:** 1Department of Orthopaedic and Trauma Surgery, Campus Bio-Medico University of Rome, 00128 Rome, Italy; g.petrucci@unicampus.it (G.P.); l.ambrosio@unicampus.it (L.A.); g.vadala@unicampus.it (G.V.); r.papalia@unicampus.it (R.P.); denaro@unicampus.it (V.D.); 2Directorate of Communication and International Affairs, Ministry of Health, 00144 Rome, Italy; s.iavicoli@inail.it

**Keywords:** COVID-19, low back pain, pandemic, lockdown experience, pain management, prevalence, physical activity, remote working

## Abstract

In March 2019, the World Health Organization (WHO) recognized the COVID-19 pandemic as a global issue. To reduce the spread of this disease, health safety pathways were implemented worldwide. These extraordinary measures changed people’s lifestyles, e.g., by being forced to isolate, and in many cases, to work remotely from home. Low back pain (LBP), the most common cause of disability worldwide, is often a symptom of COVID-19. Moreover, it is often associated with different lifestyle features (type of job, physical activity, body weight). Therefore, the purpose of this systematic review and meta-analysis was to estimate the effect of the COVID-19 lockdown on LBP intensity and prevalence compared with LBP rates before the pandemic. A systematic search was performed on Scopus, PubMed, and Cochrane Central. Overall, eight studies with 2365 patients were included in the analysis. We used the Joanna Briggs Institute (JBI) critical appraisal tool to evaluate the risk of bias: six studies (75%) were at moderate risk of bias and two studies (25%) were at low risk of bias. These studies showed an increase in both the prevalence and intensity of LBP during the COVID-19 lockdown.

## 1. Introduction

In March 2019, the World Health Organization (WHO) declared the coronavirus disease 2019 (COVID-19) a global pandemic. COVID-19 is a multisystemic syndrome with a predominant respiratory involvement caused by severe acute respiratory syndrome coronavirus-2 (SARS-CoV-2). The spreading of this virus is favored by close contact between individuals through respiratory droplets and aerosol particles [1]. Therefore, public health safety pathways have been implemented worldwide to prevent the rapid diffusion of the disease [2]. These extraordinary measures were based on social distancing, wearing masks, and additional restrictions [3,4]. This state of emergency has involved changes in the daily life of people with a significant impact on the psychological, social, and physical spheres of the population [5,6]. These measures have led to significant lifestyle changes. Indeed, people were forced to stay at home, reducing face-to-face interactions. Therefore, decreased physical activity has occurred [7], with negative consequences on patients suffering from chronic diseases and worsening the health conditions of the general population [8]. Indeed, physical activity is considered efficacious in preventing and reducing low back pain (LBP) [9,10,11,12,13]. LBP is a common condition, and most of the people suffering from this symptom tend to experience recurring episodes [14,15,16]. Indeed, LBP impacts on patients of all ages and may be persistent, eventually becoming disabling [17]. Globally, the years lived with disability originated by LBP have increased by 54% in the last 25 years [17]. LBP can be associated with physically demanding jobs [18], but it is also related to sedentary occupations [19], lower levels of physical activity, and obesity [20,21]. In a survey conducted in November 2020 in Japan, the most common population disturbance during the pandemic was indeed LBP [22]. Thus, remote working from home, reduced physical activity, and possible weight gain may have exacerbated LBP [23]. On the contrary, abstention from heavy physical work and reducing work stress may mitigate LBP manifestations. Therefore, it is unclear how the number of different factors related to the quarantine may influence the several outcomes of patients suffering from LBP. The purpose of this systematic review and meta-analysis was to estimate the effect of COVID-19 lockdown on LBP intensity and prevalence compared with the LBP rates before the pandemic.

## 2. Materials and Methods

This manuscript was elaborated following the Preferred Reporting Items for Systematic Reviews and Meta-Analysis (PRISMA) guidelines [24]. This study evaluated the clinical outcomes and prevalence of LBP before and during the COVID-19 pandemic.

### 2.1. Information Sources and Search

A systematic literature search using the string “(COVID-19) AND (low back pain)” was performed using the following databases: Scopus, Cochrane Central, and PubMed–Medline. The last search was performed on 10 November 2021. Two independent reviewers (G.F.P. and G.P.) checked the reference lists of included randomized controlled trials (RCTs) to extract additional eligible studies and, after removing duplicates, evaluated the abstracts of the selected studies. Divergences of opinion were discussed with a third review author (F.R.). Finally, the full articles were read by two review authors (G.F.P. and G.P.) to select the included studies for this systematic review and meta-analysis.

### 2.2. Inclusion Criteria

We included prospective (PS), retrospective (RS), pre-post (PP), and cross-sectional (CS) studies written in English that assessed clinical and prevalence outcomes of nonspecific LBP in a population before the pandemic due to COVID-19 and during the lockdown. We excluded studies that evaluated clinical presentations of pain that were not specific for LBP or analyzed the effect of physical activity or social isolation on pain feeling, as well as studies presenting LBP as a symptom associated with other diseases (e.g., endometriosis, cancer, trauma, systemic diseases, infection etc.).

### 2.3. Data Collection, Analysis, and Outcomes

Two reviewers (G.F.P. and G.P.) independently extracted the following data: author, year of publication, country, type of study, and the number of participants in the study group, with mean age and sex.

### 2.4. Risk of Bias Assessment

The risk of bias of the included studies was assessed using the checklist for prevalence studies of the Critical Appraisal tools developed in Joanna Briggs Institute (JBI) [25]. Two independent reviewers (G.F.P. and G.P.) performed the evaluation, and possible differences in the assessment were checked by a third reviewer (F.R.). Each item is composed of a question, for a total of nine questions. The answers were yes, no, unclear, or not applicable for each item. Thus, the studies presented a low risk of bias with eight or nine yes answers, a moderate risk of bias in the presence of seven or six yes, and a high risk of bias if fewer than six domains had yes answers.

### 2.5. Statistical Analysis

The meta-analysis was executed using the Review Manager (RevMan) software Version 5.4.1. LBP intensity was assessed as a continuous outcome using standard mean difference (SMD) with 95% confidence intervals. Instead, LBP prevalence was presented as a dichotomous outcome with an odds ratio (OR) with 95% confidence intervals. The I^2^ test was used for the heterogeneity calculation, adopting a random-effect model in high heterogeneity with I^2^ > 55%. The statistical significance of the results was fixed at *p* < 0.05.

## 3. Results

### 3.1. Literature Search

The literature search generated 260 articles. Then, 97 records were removed due to duplicates or other reasons, and the remaining 163 papers underwent title and abstract screening. After further excluding 147 studies, 16 papers were read in full text. Subsequently, eight manuscripts were excluded for the following reasons: mainly focused on physical activity (*n* = 2) [2,26], patients with LBP in the emergency department (*n* = 1) [27], clinical presentations of pain (*n* = 1) [28], expected impact of lockdown measures on LBP (*n* = 1) [29], mainly focused on loneliness and social isolation (*n* = 2) [30,31], evaluation of an intervention for LBP (*n* = 1) [32]. Finally, eight studies were included in the systematic review and meta-analysis (Figure 1).

### 3.2. Demographic Data

The total sample in all the studies was 2365 participants (Table 1). There were two prospective studies, three cross-sectional studies, one retrospective study, and t2pre-post studies. Two studies were published in 2020, while the other six were published in 2021. The number of participants per each study ranged from 50 to 528. The mean age of patients ranged from 27 to 56 years. The percentage of women in the sample ranged from 51% to 75%. Four studies (50%) involved patients suffering from chronic LBP, one study analyzed patients who underwent spine surgery within the past year for chronic LBP, one study evaluated LBP in patients affected by COVID-19, while the remaining two studies reported the incidence of LBP in a non-specific adult population.

### 3.3. Methodological Evaluation

After application of the JBI checklist, six studies (75%) resulted at moderate risk of bias and two studies (25%) at low risk of bias (Table 2). In particular, the sample frame and the sample size were judged as appropriate in almost all studies. Nonetheless, data analysis, statistical analysis, and response rate were considered adequate in the great majority of the studies. On the contrary, in question six, where measurement and classification bias were evaluated, all studies were not considered adequate. Indeed, in this tool, the objectivity was deemed to be compromised if the authors used self-reported scales, which may over- or under-report the outcome. 

### 3.4. Effect of Intervention

The meta-analysis compared the intensity and prevalence of LBP between the period before the pandemic and during the lockdown due to COVID-19.

#### 3.4.1. Low Back Pain Intensity

Pain intensity of LBP was documented by different scales in included studies. More specifically, pain was reported through the numerical rating scale (NRS) in three studies [33,39,40], and through the visual analog scale (VAS) for LBP in three studies [34,35,36]. Pain intensity showed a significant increase during the COVID-19 lockdown compared to the pre-pandemic period (SMD −1.40, 95% CI −2.18 to −0.63, *p* = 0.0004), with high heterogeneity (I^2^ = 99%) (Figure 2).

#### 3.4.2. LBP Prevalence

The analysis of the prevalence of LBP showed a statistically significant increase during the pandemic compared to the previous period (OR 0.53, 95% CI 0.29 to 0.96, *p* = 0.04), with high heterogeneity (I^2^ = 90%) (Figure 3).

## 4. Discussion

Although the most common symptoms in patients infected with COVID-19 are represented by fever, cough, and fatigue [41], it has been shown that myalgia and other diffuse musculoskeletal pains may be initial clinical manifestations of COVID-19 and can be observed at various stages of the disease [42]. Thus, musculoskeletal discomfort represents a leading cause of disease-related disability in patients with COVID-19 infection. Şahin et al. [39] have shown worsening of neck and back pain in patients with COVID-19 infection, which can persist even after infection. Moreover, it has been shown that the rate of musculoskeletal-specific pain increased from 40.7% of patients before COVID-19 to 82.5% during COVID-19, and it remained at 55.1% after COVID-19.

This systematic review and meta-analysis aimed to estimate the effect of the COVID-19 pandemic on the intensity and prevalence of LBP in various populations. Indeed, the COVID-19 pandemic itself has precipitated an unequaled global health crisis, becoming an unprecedented worldwide health issue. The need to control the diffusion of SARS-CoV-2 has forced national and international governments to implement socioeconomic measures, including confinement, arrest of non-essential production activities, and financial resources reallocation, as well to introduce new work modalities, such as remote working and teleworking [43]. Furthermore, this has profoundly impacted the general population’s social life and physical activity, with apparent consequences related to reduced exercise and a prevalently sedentary lifestyle [44]. A recent cross-sectional study on university students [45] has demonstrated that home confinement due to the COVID-19 pandemic has significantly increased sitting time and a corresponding decrease in physical activity, negatively impacting the quality of life and life satisfaction.

LBP is a multifactorial condition affecting almost every individual at least once in a lifetime and presents a significant socioeconomic issue at both healthcare and workplace levels [46]. Currently, multietiological features of LBP have been summarized by a biopsychosocial model, in which biological factors (i.e., tissue damage and degeneration due to aging, physical overload, obesity, etc. [47]) interplay with variegated psychological factors [48], including pain, catastrophizing, negative emotional responses, pain behaviors, misperceptions about the relationship between pain, health, and work and societal obstacles, loneliness, and social isolation [49]. The latter aspect may account for a substantial part of such a condition, especially in young individuals unresponsive to conventional treatments and with peculiar psychosocial traits [50]. Due to the reasons above, the COVID-19 pandemic has exacerbated psychological distress in the general population. In a recent systematic review and meta-analysis [51], the prevalence of anxiety and depression during the pandemic was 33% and 30%, respectively, mainly affecting women, younger individuals, and those with a lower socioeconomic status. Notably, these demographic factors are also known to increase the risk of LBP and chronic pain in general [52,53,54]. A study from Fallon et al. [55] found that people affected by chronic pain disorders perceived an increased pain level during the lockdown in the United Kingdom. It has been described that chronic pain syndromes may be exacerbated by social isolation. The COVID-19 pandemic may foster the impression of being ill and amplify bodily sensations’ perception, particularly in vulnerable populations, thus impacting chronic pain experience [56]. Moreover, other aspects of the new routine imposed by the pandemic may further impact the prevalence of LBP. The sedentary lifestyle and reduction of physical activity consequent to social distancing, travel restrictions, closure of exercise facilities may provoke or worsen LBP in different ways. It is widely accepted that LBP is more prevalent among office workers. Indeed, recent studies have reported that sitting for periods longer than 7 h per day significantly increases the risk of LBP (odds ratio: 1.89) [57], even if such an association remains controversial according to other reports. Remote working has inevitably incremented sitting hours. Furthermore, the absence of ergonomic facilities and dedicated workplace settings may hinder the adoption of a healthy posture and favor the onset of musculoskeletal disorders [58]. On the other hand, it is widely accepted that physical inactivity is directly associated with an increased risk of LBP and worse psychological health [59], hence promoting the vicious biopsychosocial circle. 

A recent survey among workers was focused on occupational safety and health (OSH) aspects of telework. This study demonstrated how telework is associated to several issues, such as social isolation, difficult management of working schedules with consequences on the psychosocial sphere, and a growing prevalence of musculoskeletal disorders, especially backache and muscular pains in shoulders, neck, and upper limbs. Indeed, telework is related to extended sitting postures without providing ergonomic implements. Therefore, the development of a common and integrated approach is required to organize and manage the aspects listed above. As the study suggests, working hours should be clarified, while companies should focus attention on the psychosocial risks caused by teleworking and on the design of ergonomic workstations. Therefore, it is crucial to develop strategies that facilitate dialogue with workers and their needs [60]. 

The meta-analysis showed a statistically significant increase in LBP intensity and prevalence comparing the pre-pandemic and the pandemic period (*p* = 0.0004 and *p* = 0.04, respectively). However, both outcomes presented a high heterogeneity (I^2^ = 99% for intensity and I^2^ = 90% for prevalence). Therefore, the results showed that lockdown and isolation due to COVID-19 pandemic increased pain intensity in a population of patients suffering from chronic LBP and determined a higher LBP prevalence in a non-specific adult population. However, more studies with less heterogeneity are needed to ensure greater statistical weight. A pre-post study by Licciardone et al. [33] in a sample of 476 participants who suffered from chronic LBP proved a significant reduction of the use of non-steroidal anti-inflammatory drugs (NSAIDs) for LBP during the pandemic (*p* = 0.045). In contrast, opioid use remained unchanged (*p* > 0.99). Moreover, during the lockdown, participants presented a not clinically relevant improvement in LBP intensity (mean improvement = 0.19). Bailly et al. [34] in their cross-sectional, multicenter study enrolled 360 participants suffering from chronic LBP. They demonstrated that LBP worsened in 41.1% of patients and VAS score increased from 49.5 ± 21.6 in the pre-pandemic period to 53.5 ± 22.4 during the pandemic (*p* < 0.001). On the other hand, in a prospective study conducted by Amelot et al. [35] involving 50 patients who suffered from chronic LBP, it was shown that during lockdown pain intensity improved in 36% of patients, while worsening in 28% of patients. Furthermore, the consumption of analgesics increased for 50% of patients, while it decreased for 30% of patients. In their prospective and cross-sectional study, Şan et al. [36] enrolled 145 patients who underwent spine surgery in the previous year. They showed a significant increase in the mean VAS score during the COVID-19 pandemic due to the social isolation from 4.10 ± 0.15 in the pre-pandemic period to 6.39 ± 0.30 during the pandemic (*p* = 0.000). Moreover, they demonstrated a significant increase in analgesic use, which ranged from 2.96 ± 0.33 drugs per week used in the pre-pandemic period to 5.37 ± 0.48 during the pandemic (*p* = 0.000). In their study, Abbas et al. [37] assessed LBP prevalence among 164 physiotherapy students during the COVID-19 pandemic. They did not show significant differences in LBP prevalence between the lockdown period and the 12 months. Šagát et al. [38] evaluated the effect of COVID-19 quarantine on LBP intensity and prevalence with a cross-sectional study in 463 adults residing in Riyadh. They showed that the low back was also the most painful musculoskeletal area and that LBP prevalence increased from 38.8% to 43.8% from before to during the lockdown. Finally, a pre-post study by Licciardone et al. [40] showed no significant differences in the use of NSAIDs or opioids for chronic LBP during the quarantine among 528 participants. Furthermore, they proved that the mean change of NRS from before to after the lockdown was −0.08 (95% CI, 0.21 to 0.06). The overall quality of the included studies was sufficient, presenting all a low or moderate risk of bias. 

One of the main limitations of the study was related to the presence of observer-reported or self-reported tools for the assessment of outcomes in all included articles, which may be a significant source of bias. Another limitation regards the paucity of the studies and depends on the recent capacity to analyze the relationship between COVID-19 lockdown and LBP. Moreover, the reported data are very heterogeneous in the assessment of the different outcomes. The baseline characteristic of the sample in the studies was represented by both patients suffering from chronic LBP and patients who developed LBP symptoms during the lockdown. Instead, the mean age of the participants was similar within all the studies, except for one study [37] that enrolled young physiotherapy students. Moreover, the studies presented a mild to moderate predominance of female sex among the participants.

## 5. Conclusions

In this systematic review and metanalysis, we have reported a significant increase in LBP prevalence and intensity during the COVID-19 pandemic compared to the prepandemic period. This may be explained by the reduced rate of physical activity and the prolonged sitting time without appropriate ergonomic supports during remote working. In addition, psychological implications of social isolation, including loneliness, pain catastrophizing, somatization, and the incremented risk of anxiety and depression, may further boost such a condition. Therefore, workplace interventions in the “home office” setting as well as psychological support for individuals affected by LBP are strongly advised as long as the COVID-19 pandemic persists.

## Figures and Tables

**Figure 1 ijerph-19-04599-f001:**
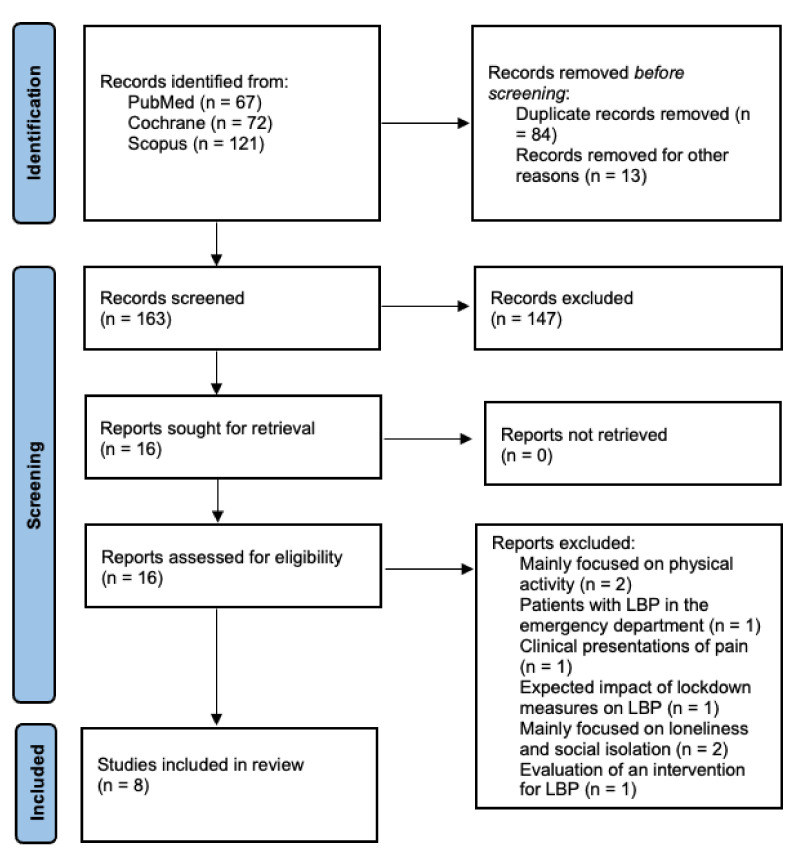
Preferred reporting items for systematic review and meta-analysis (PRISMA 2020).

**Figure 2 ijerph-19-04599-f002:**
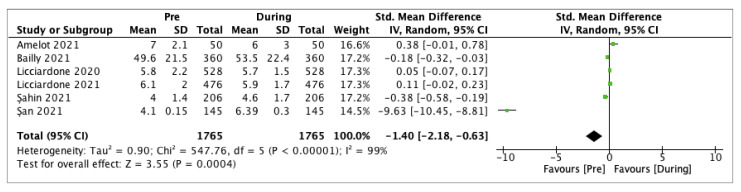
Low back pain intensity. Before versus during the pandemic [32,33,34,35,36,39].

**Figure 3 ijerph-19-04599-f003:**
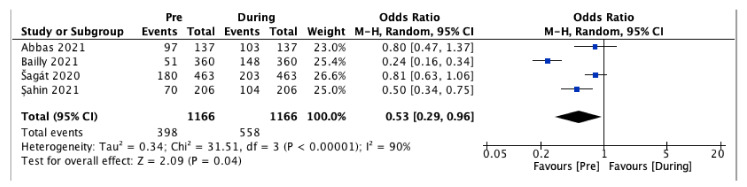
Low back pain prevalence. Before versus during the pandemic [34,36,37,38].

**Table 1 ijerph-19-04599-t001:** Main characteristics of the included studies and samples.

Author	Year	Country	Type of Study	Study Group			Population
				N.	Age	Sex	
Licciardone et al. [33]	2021	USA	PP	476	54 ± 13.2	26.7% M 73.3% F	Patients suffering from chronic LBP
Bailly et al. [34]	2021	France	CS	360	52.1 ± 13.4	41.4% M 58.6% F	Patients suffering from chronic LBP
Amelot et al. [35]	2021	France	PS	50	52.6	48% M 52% F	Patients suffering from chronic LBP
Şan et al. [36]	2021	Turkey	PS	145	54.78 ± 1.08	N.R.	Patients who underwent spine surgery within the past year for chronic LBP
Abbas et al. [37]	2021	Israel	CS	137	27 ± 3	42% M 58% F	Physiotherapy students
Šagát et al. [38]	2020	Saudi Arabia	CS	463	18–34 (*n* = 252) 35–49 (*n* = 166) 50–64 (*n* = 45)	N.R.	Non-specific adult population
Şahin et al. [39]	2021	Turkey	RS	206	56.24 ± 16.99	49% M 51% F	Patients affected by COVID-19
Licciardone et al. [40]	2020	USA	PP	528	53.9 ± 13.0	25.9% M 74.9% F	Patients suffering from chronic LBP

Abbreviations: COVID-19 = coronavirus disease 2019; CS = cross-sectional study; LBP = low back pain; PP = pre-post study; PS = prospective study; RS = retrospective study.

**Table 2 ijerph-19-04599-t002:** JBI checklist for prevalence studies.

	Licciardone (2021)	Bailly	Amelot	Şan	Abbas	Šagát	Şahin	Licciardone (2020)
Was the sample frame appropriate to address the target population?	Yes	Yes	Yes	Yes	Yes	Unclear	Yes	Yes
2. Were study participants sampled in an appropriate way?	Yes	Yes	Yes	Yes	Yes	Yes	Yes	Yes
3. Was the sample size adequate?	Yes	Yes	No	Yes	Yes	Yes	Yes	Yes
4. Were the study subjects and the setting described in detail?	Yes	Yes	Yes	Yes	Yes	Yes	Yes	Yes
5. Was the data analysis conducted with sufficient coverage of the identified sample?	Yes	Yes	Yes	Unclear	Yes	Yes	Unclear	Yes
6. Were valid methods used for the identification of the condition?	No	No	No	No	No	No	No	No
7. Was the condition measured in a standard, reliable way for all participants?	Yes	Unclear	Yes	Yes	Unclear	Yes	Yes	Yes
8. Was there appropriate statistical analysis?	Yes	Yes	Yes	Yes	Yes	Yes	Yes	Yes
9. Was the response rate adequate, and if not, was the low response rate managed appropriately?	Yes	Yes	Yes	Yes	Yes	Yes	Yes	Yes
	Low	Moderate	Moderate	Moderate	Moderate	Moderate	Moderate	Low

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
