# Peer review of "COVID-19 Pandemic Increases the Impact of Low Back Pain: A Systematic Review and Metanalysis"

_ijerph, 2022, doi:10.3390/ijerph19084599_

Round 1

Reviewer 1 Report

The authors have done a good job. 

I suggest adding limitations of this study

Author Response

We would like to thank the reviewers for their helpful comments and their suggestions for improving our manuscript.  We have carefully revised our manuscript accordingly and hope that the changes are acceptable for publication. The changes made are itemized below with our comments (dark blue text) to the reviewer’s suggestions.

Reviewer #1

The authors have done a good job. 

I suggest adding limitations of this study

Author response

Thank you very much for your appreciation of our manuscript. Limitations of the study are widely discussed in lines 252-262.

Reviewer 2 Report

Review Letters

Dear Authors

The title of this study seems to be consistent with the ijerph. I think this paper will be better if some minor and major points are corrected.

Minor points

Line 15: Low Back pain → Low back pain

Line 21: When using an abbreviation for the first time, you must first provide the full name, use the abbreviation in parentheses, and then apply it. Ex) Joanna Briggs Institute (JBI)

Line 29: Covid-19 → COVID-19

Line 40: “13]2/24/2022 8:55:00 PM” must be deleted.

Line 92: In ijerph, the statistic symbol 'p' must be written in italics. The same should be done in all cases presented below.

Sincerely,

Author Response

We would like to thank the reviewers for their helpful comments and their suggestions for improving our manuscript.  We have carefully revised our manuscript accordingly and hope that the changes are acceptable for publication. The changes made are itemized below with our comments (dark blue text) to the reviewer’s suggestions.

Reviewer #2

Dear Authors

The title of this study seems to be consistent with the ijerph. I think this paper will be better if some minor and major points are corrected.

Minor points

Line 15: Low Back pain → Low back pain

Line 21: When using an abbreviation for the first time, you must first provide the full name, use the abbreviation in parentheses, and then apply it. Ex) Joanna Briggs Institute (JBI)

Line 29: Covid-19 → COVID-19

Line 40: “13]2/24/2022 8:55:00 PM” must be deleted.

Line 92: In ijerph, the statistic symbol 'p' must be written in italics. The same should be done in all cases presented below.

Author response

Thank you very much for your comments. We have made all the proposed corrections.

Reviewer 3 Report

Please, include the references in green (archive attached).

It is a well written paper, however, it can be improved.

Some previous conditions, such as systemic sclerosis (and other rheumatologic diseases), may be cause of low back pain and should be mentioned.

Weerakoon A, Sharp D, Chapman J, Clunie G. Lumbar canal spinal stenosis due to axial skeletal calcinosis and heterotopic ossification in limited cutaneous systemic sclerosis: successful spinal decompression. Rheumatology (Oxford). 2011 Nov;50(11):2144-6.

Also, patients with systemic sclerosis are a great challenge for the physician to achieve an effective protective strategy or, when infected, to optimise a real-time treatment as suggested by the rapidly evolving guidelines.  

Systemic sclerosis and the COVID-19 pandemic: World Scleroderma Foundation preliminary advice for patient management. Ann Rheum Dis.

Besides, chest CT findings of Covid-19 may overlap with systemic sclerosis.

Mariano RZ, Rio APTD, Reis F. Covid-19 overlapping with systemic sclerosis. Rev Soc Bras Med Trop. 2020 

Was there appropriate statistical analysis? The authors should detail the statistical analysis.

Would it be possible to draw a control group from the low back pain database on sex- and age-matched patients ?

Please state if the data of these studies are from a tertiary referral Hospital?  What are the levels of the Hospitals of these studies? Referral bias?

Due to the reasons above, the COVID-19 177 pandemic has strenuously contributed to increased psychological distress in the general 178 population. In a recent systematic review and meta-analysis [ in the general 178 population. In a recent systematic review and meta-analysis [lines 177-179, discussion), please, modify: Due to the reasons above, the COVID-19 177 pandemic has exarcebated psychological distress in the general 178 population. In a recent systematic review and meta-analysis [

and a growing prevalence of musculoskeletal disorders (lines 204-205) please, detail these musculoskeletal disorders.

develop policies (line 210) develop strategies.

before the 261 pandemic in the prepandemic period

Did theses studies evaluate people who live alone?

Imaging evaluation (MRI) was performed in any of these studies?

Author Response

We would like to thank the reviewers for their helpful comments and their suggestions for improving our manuscript.  We have carefully revised our manuscript accordingly and hope that the changes are acceptable for publication. The changes made are itemized below with our comments (dark blue text) to the reviewer’s suggestions.

Reviewer #3

Please, include the references in green (archive attached).

It is a well written paper, however, it can be improved.

Some previous conditions, such as systemic sclerosis (and other rheumatologic diseases), may be cause of low back pain and should be mentioned.

Weerakoon A, Sharp D, Chapman J, Clunie G. Lumbar canal spinal stenosis due to axial skeletal calcinosis and heterotopic ossification in limited cutaneous systemic sclerosis: successful spinal decompression. Rheumatology (Oxford). 2011 Nov;50(11):2144-6.

Also, patients with systemic sclerosis are a great challenge for the physician to achieve an effective protective strategy or, when infected, to optimise a real-time treatment as suggested by the rapidly evolving guidelines. 

Systemic sclerosis and the COVID-19 pandemic: World Scleroderma Foundation preliminary advice for patient management. Ann Rheum Dis.

Besides, chest CT findings of Covid-19 may overlap with systemic sclerosis.

Mariano RZ, Rio APTD, Reis F. Covid-19 overlapping with systemic sclerosis. Rev Soc Bras Med Trop. 2020

Author response

Thank you very much for your suggestions. We specifically focused our systematic review and metanalysis on the prevalence and intensity of LBP in the general population before and during the pandemic. LBP is a ubiquitous symptom and may be caused by several disorders. However, we aimed to focus on nonspecific LBP only, as discussing all the numerous additional causes of LBP as well as their radiological manifestations in COVID-19 patients (as suggested) will let us fall outside our research question. This has been now better specified in Section 2.2.

Was there appropriate statistical analysis? The authors should detail the statistical analysis.

Author response

Thank you very much for your comment. We believe that the statistical analysis is adequate, and it is reported in the sections 2.5 and 3.4.

Would it be possible to draw a control group from the low back pain database on sex- and age-matched patients?

Author response

Thank you for your interesting comment. Unfortunately, the included studies did not perform subgroup analysis for sex and age.

Please state if the data of these studies are from a tertiary referral Hospital?  What are the levels of the Hospitals of these studies? Referral bias?

Author response

Thank you for the appropriate comment. Only in the study by Åžahin et al. and Amelot et al. the data was collected in Hospitals. In all the other studies, prevalence and incidence of LBP were collected among non-hospitalized population.

Due to the reasons above, the COVID-19 177 pandemic has strenuously contributed to increased psychological distress in the general 178 population. In a recent systematic review and meta-analysis [ in the general 178 population. In a recent systematic review and meta-analysis [lines 177-179, discussion), please, modify: Due to the reasons above, the COVID-19 177 pandemic has exarcebated psychological distress in the general 178 population. In a recent systematic review and meta-analysis [

Author response

Thank you very much for your comments. We have made all the proposed corrections.

and a growing prevalence of musculoskeletal disorders (lines 204-205) please, detail these musculoskeletal disorders.

Author response

Thank you very your comment. We inserted the musculoskeletal disorders.

develop policies (line 210) develop strategies.

before the 261 pandemic in the prepandemic period

Author response

Thank you very much for your comments. We have made all the proposed corrections.

Did theses studies evaluate people who live alone?

Author response

Thank you for your appropriate comment. Only one study (Ayça Uran Åžan et al.) evaluated the role of social isolation on LBP during the COVID-19 pandemic. However, the remaining studies did not evaluate people who live alone.

Imaging evaluation (MRI) was performed in any of these studies?

Author response

Thank you for your interesting comment. In the included studies MRI was not performed, while LBP was assessed through clinical examination and the reported outcomes.

Round 2

Reviewer 1 Report

The authors have addresses the concerns of the reviewers.

Reviewer 3 Report

the reviewed form of the paper followed my suggestions